# Putative Novel Atypical BTV Serotype ‘36’ Identified in Small Ruminants in Switzerland

**DOI:** 10.3390/v13050721

**Published:** 2021-04-21

**Authors:** Christina Ries, Andrea Vögtlin, Daniela Hüssy, Tabea Jandt, Hansjörg Gobet, Monika Hilbe, Carole Burgener, Luzia Schweizer, Stephanie Häfliger-Speiser, Martin Beer, Bernd Hoffmann

**Affiliations:** 1Institute of Diagnostic Virology, Friedrich-Loeffler-Institut, Südufer 10, 17493 Greifswald-Insel Riems, Germany; Christina.Ries@fli.de (C.R.); martin.beer@fli.de (M.B.); 2Institute of Virology and Immunology (IVI), Mittelhäusern, Switzerland and Department of Infectious Diseases and Pathobiology, Vetsuisse Faculty, University of Bern, 3012 Bern, Switzerland; Andrea.Voegtlin@ivi.admin.ch (A.V.); Daniela.Huessy@ivi.admin.ch (D.H.); Tabea.Jandt@gmx.ch (T.J.); hansjoerg.gobet@ivi.admin.ch (H.G.); 3Institute of Veterinary Pathology, Vetsuisse Faculty, University of Zürich, 8057 Zürich, Switzerland; hilbe@vetpath.uzh.ch (M.H.); carole.burgener@uzh.ch (C.B.); 4Tierarztpraxis Kreuzberg, 9473 Gams, Switzerland; schweizer@meintierarzt.ch; 5Beratungs-und Gesundheitsdienst für Kleinwiederkäuer BGK, 3362 Niederönz, Switzerland; stephanie.haefliger@caprovis.ch

**Keywords:** bluetongue virus, BTV, serotype 36, atypical BTV, small ruminant, novel serotype, goat

## Abstract

We identified a putative novel atypical BTV serotype ‘36’ in Swiss goat flocks. In the initial flock clinical signs consisting of multifocal purulent dermatitis, facial oedema and fever were observed. Following BTV detection by RT-qPCR, serotyping identified BTV-25 and also a putative novel BTV serotype in several of the affected goats. We successfully propagated the so-called “BTV-36-CH2019” strain in cell culture, developed a specific RT-qPCR targeting Segment 2, and generated the full genome by high-throughput sequencing. Furthermore, we experimentally infected goats with BTV-36-CH2019. Regularly, EDTA blood, serum and diverse swab samples were collected. Throughout the experiment, neither fever nor clinical disease was observed in any of the inoculated goats. Four goats developed BTV viremia, whereas one inoculated goat and the two contact animals remained negative. No viral RNA was detected in the swab samples collected from nose, mouth, eye, and rectum, and thus the experimental infection of goats using this novel BTV serotype delivered no indications for any clinical symptoms or vector-free virus transmission pathways. The subclinical infection of the four goats is in accordance with the reports for other atypical BTVs. However, the clinical signs of the initial goat flock did most likely not result from infection with the novel BTV-36-CH0219.

## 1. Introduction

Bluetongue virus (BTV; genus *Orbivirus*) is a non-enveloped double-stranded RNA virus with 10 segments causing bluetongue disease (BT) in ruminants [1]. Segment 2 encoding for virus protein 2 plays the major role for serotype specificity [2]. There is a constantly growing number of atypical serotypes with different viral characteristics which are genetically distinct from the 24 classical serotypes. Difficulties in the proper serotyping of these novel atypical strains allowed for their classification on the molecular level as ‘putative novel (atypical) serotypes’ only [3]. The classical serotypes 1-24 are listed as notifiable disease by the OIE, whereas the novel atypical serotypes do not need to be controlled [4]. The natural route of infection of the arbovirus BTV is transdermal infection via blood feeding *Culicoides* midges as competent vectors [5]. Nevertheless, infectious blood entering via wounds or syringes was also described as a possible source of BTV infection [6]. After the first replication cycle in leukocytes in the regional lymph node, the virus disseminates to secondary sites of replication such as lungs, lymph nodes, spleen and skin [7]. Virus replication occurs in mononuclear phagocytic, dendritic and endothelial cells; thus, bluetongue disease is characterized by vascular injury with haemorrhages of mucous membranes and skin, tissue infarction, widespread oedema, pyrexia, apathy, hyperaemia, lameness, respiratory disorders and oral erosions and ulcers [5,8,9]. The most clinically affected and highly susceptible animal species for classical BTV serotypes 1–24 are sheep, but the severity of disease strongly depends on breed, virus strain, herd immunity or genetics of the individual animals [10]. Nevertheless, many other species of domestic and wild ruminants are susceptible, including cattle, but with much lower morbidity and mortality rates [11]. 

For a long time, the distribution of BTV was limited to tropical and subtropical areas. Before 1998, BTV occurred only sporadically in Europe [12]. However, since 1998 due to increased global travel, trade and climate change, *Culicoides* midges have spread northwards and the endemic circulation of several BTV serotypes in Europe began [10,12]. In 2002, Switzerland agreed to follow the EU regulations regarding animal health and disease control [13] and in 2003 launched an early warning system against BTV including vector surveillance [14]. Abundant *Culicoides* species were found in Switzerland and sentinel herd surveillance was established in 2004 [14]. In October 2007, the first BTV-8 cases emerged in Switzerland near Basel [13,15]. Swiss bluetongue surveillance from 2007 to 2008 was a combination of monthly bulk milk testing in zones of higher risk areas and passive clinical surveillance. From July 2008 until spring 2010, BTV vaccination was obligatory in Switzerland for cattle, sheep and goats and bulk milk surveillance became irrelevant. A Swiss pathogenicity study for the relevant sheep breeds in Switzerland was conducted in 2010 revealing that all Swiss sheep breeds were highly susceptible to BTV-8 infection as demonstrated by the clinical-pathological picture and viremia during experimental infection [16]. Interesting findings resulted from a survey study of the free-ranging Swiss wildlife population, where 0.5% of 1898 animals tested were seropositive for BTV-8 specific antibodies and 0.3% were positive for BTV-8 RNA. The last BTV-8 case in Switzerland was detected in March 2010, and in 2012 Switzerland officially reached the disease-free status for the next 4 years [13]. Since November 2017, Switzerland has been declared a BTV-8 restriction zone due to two BTV-8 positive tested farms. In 2018, 75 BTV-8 outbreaks were reported in Switzerland and in 2019 still 53 cases. In 2020, only four BTV-8 cases were reported, with the last BTV-8 case report in November 2020 [17,18,19,20]. 

In the course of the intensive BTV surveillance and monitoring in Switzerland, a novel bluetongue serotype was discovered in 2008 and marked the beginning of several genetically closely related novel BTV serotypes discovered all over the world [15]. The first detected atypical BTV serotype was found to circulate in clinically healthy goats in two goat flocks in the canton of St. Gallen and was named “Toggenburg virus” or “TOV” after the region of its discovery. Genetic analysis revealed a clear molecular distance from the 24 classical serotypes for several segments, and the maximum sequence identities to any other BTV serotype ranged from 63% (Segment 2) to 79% (Segments 7 and 10) at the time point of discovery [21]. TOV did not show any BTV related clinical signs in field and experimentally infected goats and only mild clinical signs in experimentally infected sheep [22]. Even though the prevalence of TOV in the goat flocks was high, no cattle were found to be positive on the affected farms, leading to the conclusion that TOV is mainly a virus of small ruminants. This finding is in harmony with the later discovery of other atypical BTV serotypes mainly detected in clinically healthy goats such as BTV-26 from Kuwait (the first atypical BTV strain isolated in cell culture) [23], BTV-27 from France [24], BTV-XJ2014 from China [25] and BTV-X ITL2015 from Italy [26]. Furthermore, TOV was the first reported BTV serotype which has not been cultivated neither in cell culture nor in embryonated chicken eggs [27] followed by BTV-X ITL2015 and BTV-Z ITL2017 which so far have been non-cultivable as well [26,28]. BTV-25-GER2018 represented the first BTV-25 isolate cultivated in cell culture [29]. 

In our study, we present the discovery of two atypical BTV strains in clinically affected Swiss goat herd. In the context of diagnostic clarification, BTV positivity in RT-qPCR [15] was noticed. After serotyping, the coincident presence of two different BTV serotypes in the affected herd was ascertained, namely, BTV-25-CH2019-1 and BTV-36-CH2019. Both virus strains were successfully propagated in cell culture, and the full genome revealed that BTV-25-CH2019-1 shows high sequence identities with a BTV-25 isolate from Germany detected in 2018. In contrast, BTV-36-CH2019 is a putative novel atypical serotype and the experimental infection of goats using this novel BTV serotype delivered no indications for any enhanced virulence or any vector-free virus transmission.

## 2. Materials and Methods 

### 2.1. Sampling in Switzerland

#### 2.1.1. Goat Flock A

In total, 77 EDTA blood samples were taken from the clinical affected Goat Flock A, which was kept on an alp at an altitude of about 1839 m above sea level in the canton of St. Gallen. Goat Flock A comprised 77 animals, including 44 young animals born in 2019 and 33 adult animals.Thereof, 50 animals were showing signs of disease. In total, 5 males and 51 females were present, whereas for 21 animals the gender was not determined. The breeds included Stiegfelgeiss (1), Capra grigia (1), Saanen (1), Gämsfarbige Gebirgsziege (2), Bündner Strahlenziege (3), Appenzeller (5), Toggenburger (35), and 29 animals were mixed breeds. None of the animals were vaccinated against BTV.

#### 2.1.2. Goat Flocks B, C and D

For surveillance, in total 24 EDTA blood samples of 24 goats from a neighboring alpine area were taken (Flock B). Animals tested from Flock B were adult and female and belonged to six different breeds, namely, Gämsfarbige Gebirgsziege (4), Saanen (1), Appenzeller (2), Nera Verzasca (3), Bündner Strahlenziege (4) and Toggenburger (10). Furthermore, in total, 15 samples of 15 goats from Goat Flock C and 15 samples of 15 goats from Goat Flock D, both located in lowlands, were taken and analyzed. Goats tested from Flock C consisted of 12 adults and 3 young animals born in 2019, all Toggenburger breed. One male and 14 females were present in Flock C. In Flock D, 11 adult goats were female and 1 was male, all belonging to the Toggenburger (10) and Bünder Strahlenziege breeds (2). For three goats, no breed information was available. None of the animals were vaccinated against BTV.

#### 2.1.3. Diagnostic Clarifications

EDTA blood samples originating from two clinically infected goats from Flock A were sent to the Institute of Virology and Immunology (IVI) in Mittelhäusern, Switzerland, for laboratory investigation (molecular and serological). In one sample, BTV RNA was detected without clarification of the serotype. Subsequent analysis included testing of a number of other animals and revealed similar results. In order to determine the serotype with further molecular assays, sequencing and virus isolation, samples were sent to the Friedrich-Loeffler Institut (FLI) on the Isle of Riems, Germany. In parallel, five severe clincial diseased goats were euthanized and sent to the Institute of Veterinary Pathology, Vetsuisse Faculty, Zurich, for section and histology.

### 2.2. RNA Extraction and RT-qPCR

The field blood samples were processed in the IVI in Switzerland. For this purpose, viral RNA was manually extracted from the field EDTA blood samples using Trizol (Invitrogen, Carlsbad, CA, USA) according to the manufacturer’s instructions. Subsequently, samples were analyzed for the presence of BTV RNA by various RT-qPCR assays. First, the Pan-BTV-S10 RT-qPCR method [15] was used to screen the samples for detection of viral RNA from any potential BTV serotype. Subsequently, for all positive samples it was tried to identify the serotype by different serotype-specific real-time RT-qPCR assays (BTV-1, BTV-4, BTV-8, BTV-25). For BTV-1, BTV-4 and BTV-8 commercial available RT-qPCR kits (VetMAX European BTV Typing Kit of Thermo Fisher Scientific) were used, whereas for BTV-25 detection a previously published assay [29].

At the FLI in Germany, all organ samples (Skin, lung, liver, spleen, mesenteric lymphnode) were homogenized in serum-free medium using the TissueLyser II tissue homogenizer (QIAGEN, Hilden, Germany). Viral RNA of 100 µL sample material (EDTA blood; serum; nasal, ocular, oral and rectal swabs; homogenised organs or cell culture material) was extracted using the NucleoMagVET kit (Macherey-Nagel, Düren, Germany) and the half-automated KingFisher platform (King-Fisher Flex magnetic particle processor, Thermo Fisher Scientific) and eluated in 100 µL elution buffer. For control of successful RNA extraction, an internal control RNA (IC-2 RNA) was added during the extraction process [30]. The RNA was amplified using the Pan-BTV-S10-RT-qPCR recommended by the OIE [15] modified by an internal control amplification. The final composition of the RT-qPCR reactions was 1.25 μL of RNase-free water, 6.25 μL of 2× RT-PCR buffer, 0.5 μL of RT-PCR Enzyme Mix, 1 μL of primer-probe-mix-FAM, 1 μL of EGFP-mix1-HEX and 2.5 µL of the heat denatured template RNA. All RT-qPCRs were run on the CFX 96 real-time PCR cycler (Bio-Rad, Hercules, CA, USA) with the AgPath-ID™ One-Step RT-PCR Reagents of Applied Biosystems™ (Waltham, ME, USA). The temperature profile used was 10 min at 45 °C, 10 min at 95 °C followed by 42 cycles of 15 s at 95 °C, 20 s at 56 °C, and 30 s at 72 °C. Fluorescence values (FAM, HEX) were collected during the annealing step. Samples were considered positive when quantification cycle (Cq) values were <40.

For identification of the novel atypical BTV serotype, a newly designed BTV-atypS2-K10 RT-PCR for the partial sequencing of Segment 2 was applied. The forward primer BTV-atypS2-2285F (5’-AGA CAR TGG TCG RTT CCK ATG AT-3’) and the reverse primer BTV-atypS2-2883R (5’- GTC ATS AGY TCA TCG TTT CCA AA-3’) were used to amplify a 599 bp fragment. Both primers were designed by using the available Seg-2 sequences of all published atypical BTV strains and succesfully validated with different atypical serotypes including BTV-25, BTV-26 and BTV-27. For the RT-PCR the SuperScript™ III One-Step RT-PCR System with Platinum™ Taq DNA Polymerase (Fisher Scientific, Schwerte, Germany) was used. Briefly, for an RT-PCR reaction, 0.75 µL RNase-free water, 6.25 µL 2× Reaction Mix, 0.5 µL MgSO_4_, 0.5 µL Enzyme mix, 1.0 µL forward primer BTV-atypS2-2285F (10 pmol/µL) and 1.0 µL BTV-atypS2-2883R (10 pmol/µL) were mixed. The master mix of 10 µL was added to 2.5 µL of the heat-denaturated template RNA and the RT-PCR was started by using this temperature profile: 30 min at 50 °C, 2 min at 95 °C followed by 50 cycles of 15 s at 95 °C, 30 s at 56 °C, and 60 s at 72 °C. After a final elongation for 5 min at 72 °C, the successful amplification of the PCR fragment was checked by electrophoretic analysis in a 1.5% TAE gel. Positive PCR fragments of approx 599 bp were cut out, purified by using the QIAquick Gel Extraction Kit (Qiagen, Hilden, Germany) and sequenced with the identical forward and reverse primer by using Sanger sequencing standard procedure. Based on the generated partial Seg-2 sequence of the novel BTV-36-CH2019 strain, a sensitive and specific BTV-36 Mix 2 assay was designed and evaluated. The forward primer BTV-36-157F (5′- AAC GAT GAT TAT GCA AGG CAA G -3′) and the reverse primer BTV-36-283R (5’- CGC GAC TGG TAA AGT ATA TAC AAT -3’) were used in a 0.8 µM concentration, whereas the probe BTV-36-205FAM (6-FAM- ATT GGA TAC GCT TAA GAC ACA CCT GG -BHQ1) was used in a 0.2 µM concentration for preparation of the primer–probe mixture. The protocol of the BTV-36 assays was based on the published BTV-25 specific RT-qPCR assay [29] and was only adapted with the BTV-36 primer and probe. The detailed protocol is available on request. Finally, the newly developed BTV-36 Mix 2 assay and the published BTV-25 specific RT-qPCR assay [29] were applied for the testing of the collected specimens.

### 2.3. Virus Isolation in Cell Culture

For the initial virus isolation experiments of BTV-36-CH2019, we chose 8 EDTA blood samples of the field-infected Swiss goats with Cq values ranging from 26.5 to 28.7 in the Pan-BTV-S10-RT-qPCR. Blood samples were processed for the virus isolation experiments as follows: 500 µL of EDTA blood was centrifuged at 5000 rpm for 5 min. Then, the red blood cells were washed twice in 1 mL PBS and finally diluted in 500 µL PBS prior to lysis by 20 s ultrasound treatment with a frequency of 20 kHz (Sonifier 450, Branson Ultrasonics, Brookfield, USA). BHK-21 (BSR/5) cells (FLI cell culture collection number RIE0194) in T25 cm^2^ cell flasks were incubated initially for three hours at 37 °C using the Eagles cultivation medium with Earles and Hank’s salts with non-essential amino acids (FLI internal medium number ZB5d) supplemented with 10% FCS (fetal calf serum). Afterwards, the cells were inoculated with 300 µL of washed blood cells for two hours. After incubation at 37 °C, the blood inoculum was removed and flasks were refilled with medium supplemented with 10% FCS and antibiotics in double standard concentration (20,000 µg/mL Penicilin, 20,000 units/mL Streptomycin, 10 mg/mL Gentamicin, 250 µg/mL Amphotericin (Thermo Fisher Scientific, Waltham, MA, USA) and incubated for 4–5 days at 37 °C. In a next step, the cells of the infected BSR monolayer were detached by using 1 mL of trypsin and afterwards mixed with 2 mL of fresh medium supplemented with 10% FCS and antibiotics. In a next step, 2 mL of this cell-trypsin-medium suspension was transferred to a new T25 cm^2^ cell flask. Three passages were performed and the success of virus replication was confirmed by genomic load estimated by RT-qPCR. The virus stock prepared for the experimental inoculation was passaged for times on BSR cells.

The virus isolation experiments of BTV-25-CH2019-1 were performed on BSR cells with an identical procedure. We chose 4 EDTA blood samples of the field-infected Swiss goats positive in the BTV-25 specific RT-qPCR with Cq values ranging from 24.2 to 29.1.

### 2.4. Serological Analyses

#### 2.4.1. Production of a Polyclonal Antiserum

One New Zealand white rabbit was immunized with a mixture of binary ethyleneimine (BEI)-inactivated BTV-36-CH2019 full-virus cell culture material. Binary ethylenimine (BEI) was prepared freshly by cyclization of 0.1 M 2-bromoethylamine hydrobromide in 200 mM sodium hydroxide (NaOH) solution at 37 °C for 60 min [31]. Before inactivation, the BTV-36-CH2019 cell culture preparations had a titer of 10^4.8^ CCID_50_/_mL_ (cell culture infectious dose). BEI inactivation was based on a standard procedure [31] and the antigen preparations were stored at −80 °C until use. The success of the inactivation procedure was confirmed by three serial passages on cell culture and verification of the decrease of genome load by RT-qPCR. The rabbit was inoculated subcutaneously three times at two-week intervals with 1 mL of inactivated antigen mixed with 100 µL of Polygen as adjuvant (MVP Adjuvants^®^, Omaha, USA). The final serum was collected at 56 dpv (days post vaccination). The respective experimental protocols were reviewed by the state ethics commission and approved by the competent authority (State Office for Agriculture, Food Safety and Fisheries of Mecklenburg-Vorpommern, Rostock, Germany; Ref. LALLF M-V/TSD/7221.3-2-042/17).

#### 2.4.2. ELISA

At the IVI in Switzerland, the commercially available ELISA INgezim BTV DR 12.BTV.K.0 (INGENASA, Eurofins, Budapest, Hungary) targeting the VP7 was used according to the manufacturer’s protocol for analysis of the field plasma samples. Samples were considered positive when the OD value at 450 nm was higher than the cut-off (15% of positive control). Samples were considered negative if the OD value was equal or lower than the positive cut-off.

At the FLI in Germany, all serum samples from the animal experiments were screened for BTV-group-specific antibodies using a cELISA targeting the VP7 (ID Screen^®^ Bluetongue Competition, ID-Vet, France) according to the manufacturer’s instructions. The cut-off of 50% according to the manufacture’s instruction was applied. Samples with S/N ≤ 50% were considered positive, samples with >50% S/N as negative.

#### 2.4.3. Virus Neutralization Test

A virus neutralization test (VNT) was performed for the detection of serotype-specific neutralizing antibodies. BTV-36-CH2019 was used after five passages on BSR cells with a virus titer of 10^4.83^ CCID_50_/_mL_. VNTs were run with the cELISA-positive polyclonal rabbit serum, with the reference sera of classical BTV serotypes 1–24 (generated in guinea pig or rabbits) and the sera reactive against BTV-25-GER2018, BTV-26, BTV-27x, BTV-28, BTV-MNG1/2018, BTV-MNG2/2016 and BTV-MNG3/2016. Furthermore, the sera of the experimentally infected goats on 42 dpi was tested in VNT as well. Briefly, the serum was diluted in log2 steps starting from 1:10 to 1:280 and titrated against 100 CCID_50_ of BTV-36-CH2019 per well. Plates were incubated for 1 h at 37 °C before overnight incubation at 4 °C. The following day, 100 µL of a BSR cell suspension of approx. 30,000 cells/100 µL were added per well. After incubation for 4 days at 37 °C, all wells were scored for a cytopathic effect (CPE). The neutralization titer was determined as the dilution of serum giving 100% neutralization. The calculations according to the Spearman and Kärber method were used.

### 2.5. Experimental Infection of Goats

Seven male, approximately 6-month-old BTV seronegative Thuringian goats were kept in the vector-free high containment buildings of the FLI, Isle of Riems. Five goats (#12, #13, #14, #15, #17) were inoculated subcutaneously on the shoulder with 2 mL each of BTV-36-CH2019-infected BSR cell culture material in passage four with an infectious titer of 10^5.0^ CCID_50_/_mL_. The two remaining goats (#18, #19) were kept as control goats for horizontal transmission in the same enclosure with direct contact. All seven goats were kept until 46 dpi and monitored daily for clinical signs with the previously described clinical score [3]. EDTA blood and serum samples, as well as nasal, ocular, oral and rectal swabs were taken regularly on days 0, 3, 5, 7, 10, 12, 14, 17, 21, 28, 35 and 42 days post infection (dpi) of all goats. The respective experimental protocols were reviewed by the state ethics commission and approved by the competent authority (State Office for Agriculture, Food Safety and Fisheries of Mecklenburg-Vorpommern, Rostock, Germany; Ref. LALLF M-V/TSD/7221.3-1-008/19). 

### 2.6. Sequence Analysis

For BTV-36-CH2019, the cell culture isolate (BSR passage 3 with Ct-value of 11.7) of a clinical diseased goat of Flock A was chosen for sequencing, as well as for BTV-25-CH2019 (BSR4 with a Ct-value of 14.3). The sequences of the ten segments of BTV-36-CH2019 and BTV-25-CH2019-1 were generated using the HTS-SISPA technology [32] on the Illumina platform. Samples for sequencing were processed as described previously [29]. The amplified and purified cDNA was sent to Eurofins Genomics (Ebersberg, Germany) for sequencing on an Hiseq Illumina platform. Raw data as fastq files were trimmed and assembled by mapping to the BTV-25 TOV reference sequences with the following accession numbers: GQ982522 (Seg-1), EU839840 (Seg-2), GQ982523 (Seg-3), GQ982524 (Seg-4), EU839841 (Seg-5), EU839842 (Seg-6), EU839843 (Seg-7), EU839844 (Seg-8), EU839845 (Seg-9), EU839846 (Seg-10) using Geneious software v2019.2.3 (Biomatters Ltd., Auckland, New Zealand). The BTV-36-CH2019 sequences obtained in this study were submitted to the European Nucleotide Archive (ENA) with the following accession numbers: Seg-1: LR993239, Seg-2: LR993240, Seg-3: LR993241, Seg-4: LR993242, Seg-5: LR993243, Seg-6: LR993244, Seg-7: LR993245, Seg-8: LR993246, Seg-9: LR993247, and Seg-10: LR993248. The BTV-25-CH2019-1 sequences were submitted to ENA as well, with the following accession numbers: Seg-1: LR993249, Seg-2: LR993250, Seg-3: LR993251, Seg-4: LR993252, Seg-5: LR993253, Seg-6: LR993254, Seg-7: LR993255, Seg-8: LR993256, Seg-9: LR993257, Seg-10: LR993258. For phylogenetic analyses, a multiple alignment of BTV sequences was performed by using the MAFFT alignment feature in the Geneious software. We included the same BTV strains representing the known BTV serotypes as used in the publication of BTV-X- ITL2015 and BTV-25-GER2018 [26,29]. Phylogenetic trees of each of the 10 segments of the novel Swiss strains BTV-36-CH2019 and BTV-25-CH2019-1 were created with MegaX using the genetic distinction model Tamura–Nei and tree-built method Maximum likelihood [33]. To assess the robustness of individual nodes on the phylogenetic trees, we performed a bootstrap analysis with 100 replications. Furthermore, the consensus sequences of each of the 10 segments of BTV-36-CH2019 were determined by BLAST (NCBI) analysis against the nucleotide/amino acid (nt/aa) databases to identify the nearest molecular neighbors.

## 3. Results

### 3.1. Sampling in Switzerland

#### 3.1.1. Clinical Findings 

Clinical signs including sudden high fever and head oedema were observed in 41 goats originating from Flock A, 34 animals were clinically healthy, and for two animals, no information was available. All goats from Flocks B, C, and D were healthy. 

#### 3.1.2. Pathological Findings

Macroscopically, severe skin alterations could be seen on the ears, the bridge of the nose, the shoulder blades, and along the spine. The skin was reddened, alopecic with severe crusting as well as with serous exudation. In some areas, the skin and subcutaneous tissue were severely necrotic and had started to peel off. Histologically, a severe multifocal necrosuppurative dermatitis with orthokeratotic hyperkeratosis and acanthosis was visible. Additionally, granulation tissue formation was detected, suggesting that these findings are not entirely acute, rather subacute. Furthermore, severe coagulation necrosis and fibrinoid vascular changes were observed in two of the five examined goats.

### 3.2. BTV RT-qPCR

In total, 131 EDTA blood samples were analyzed for the presence of BTV RNA using the Pan-BTV-S10 RT-qPCR assay. From Flock A, 64 of the 77 samples revealed a positive Pan-BTV-S10 RT-qPCR, whereas 13 yielded a negative result. From that 64 BTV RNA positive samples of Flock A, 33 were positive in the BTV-25 specific assay and 54 were positive in the BTV-36 specific assay. However, 27 samples of Flock A were positive in both assays.

From Flock B, BTV RNA was detected in only 3 of the 24 samples, whereas in Flock C in 1 out of 15 samples and in Flock D 3 out of 15 samples were detected. In one of the three BTV-positive samples from Flock B, RNA from both, BTV-25 and BTV-36, was detected, whereas one sample contained only BTV-36 RNA and one sample was undeterminable. 

All four BTV-positive samples originating from Flocks C and D in the lowland were positive for BTV-25 RNA only.

### 3.3. Virus Isolation In Vitro

From five of the eight chosen field blood samples, the BTV-36-CH2019 strain was isolated successfully. After the second and third cell culture passage, a CPE was observed and successful propagation of the virus was confirmed with decreasing Cq values in the RT-qPCR in higher cell passages. A maximum titer of 10^5.0^ CCID_50_/_mL_ of the infected cell culture material was achieved with the Cq value of 11.7 in the Pan-BTV-S10 RT-qPCR.

From three of the four selected field blood samples, only positive for the BTV-25 genome, the according virus was isolated successfully. A substantial CPE was observed in the third cell culture passage. The successful propagation of the BTV-25-CH2019 isolates was also confirmed by RT-qPCR.

### 3.4. Serological Investigations

#### 3.4.1. Polyclonal Antiserum

A polyclonal BTV-36-CH2019 antiserum was generated from an immunized New Zealand white rabbit. The final rabbit serum collected on day 56 post vaccination was positive in cELISA for group-specific BTV antibodies (log2 cELISA titer up to 1:8). Nevertheless, no neutralizing antibodies were detected for the rabbit sera due to an incomplete neutralization pattern in the VNT.

#### 3.4.2. ELISA Results of Field Sera

In 52 out of 77 plasma samples collected from Flock A, antibodies against BTV were detected and 25 were negative for BTV antibodies in the Ingenasa ELISA. A total of 43 goats were positive in the ELISA and in the RT-qPCR, whereas 21 goats were negative in the ELISA, but positive in the RT-qPCR. In total, four animals were negative for both BTV antibodies and RNA. In 16 out of 24 samples from Flock B, antibodies were present. Concerning Flocks C and D in the lowland, 11 and 12 samples out of 15 revealed a positive result in the serology. All animals in Flocks C and D which tested positive for viral RNA were positive for the presence of antibodies against BTV in the ELISA.

#### 3.4.3. Virus Neutralization

The virus neutralization test of the cELISA-positive rabbit serum led to incomplete neutralization of BTV-36-CH2019. At the 1:10 dilution step of the serum dilution, a partial CPE was observed in a part of the cell monolayer in four of the in total six wells, whereas in two wells complete neutralization was observed. The CPE increased with the higher dilution steps until a 75–100% CPE was seen at dilution step 1:80. The BTV-36-CH2019 goat sera of the experimentally infected goats, the reference sera BTV 1 to 24, as well as the sera of BTV-25-GER2018, BTV-26, BTV-27x, BTV-28, BTV-MNG1/2018, BTV-MNG2/2016 and BTV-MNG3/2016 did not neutralize BTV-36-CH2019 with a 100% CPE already in the 1:10 dilution. Interestingly, the BTV-MNG1/2018 serum, produced in a rabbit as well, showed a nearly similar partial neutralization capacity as the BTV-36-CH2019 serum.

### 3.5. Experimental Inoculation of Goats

No clinical signs and fever were observed during the whole experiment in any of the goats. The RT-qPCR analysis of the EDTA blood samples revealed that goats #13 and #14 were positive for BTV genome starting on 10 dpi, whereas for goat #15 on 12 dpi, followed by goat #12 on 17 dpi. However, goat #17 did not develop viremia during the whole length of the experiment despite subcutaneous infection. The two control goats #18 and #19 remained negative in the RT-qPCR throughout the animal trial. Furthermore, infected and control goats stayed clearly negative in the cELISA during the animal experiment starting on 0 dpi until 42 dpi. The results of the RT-qPCR of the EDTA blood and the cELISA results of the serum are shown in Figure 1.

All oral, nasal, ocular and rectal swabs were tested negative for BTV genomes at all sampling time points with the exception of one nasal swab of goat #14 on 21 dpi (Cq value 33.1). The organs of all goats did not show any pathological abnormalities. The lung, liver and spleen of goats #12, #13, #14 and #15 were positive in the RT-qPCR with Cq values ranging from 31.1 to 34.4, whereas both skin samples and lymph node samples of the respective goats were tested negative for BTV genomes. All organs of goats #17, #18 and #19 were negative in the RT-qPCR. Results of the organs on 46 dpi and the EDTA blood on 42 dpi are shown in Table 1. 

### 3.6. Sequence Analysis

The sequences of all 10 segments of both BTV-36-CH2019 and BTV-25-CH20019-1 were generated and used for phylogenetic analyses (Figure 2) including BTV strains representing the known BTV serotypes [26,34]. BTV-25-CH20019-1 resembles in all segments BTV-25-GER2018, which is in line with the phylogenetic trees. Segment 2 of BTV-25-CH20019-1 showed 98.80% identity in nt with BTV-25-GER20018, and 89.78% identity in aa with TOV. Thus, BTV-25-CH2019-1 is part of serotype 25.

For BTV-36-CH20019, the Segment 2 nt identities to BTV strains used in the phylogenetic tree varied from 40.0% (BTV-15) up to 58.9% (BTV-10) for the classical serotypes 1-24. The identity of the atypical serotypes started from 55.7% (BTV-28) up to 62.5% for BTV-MNG2/2016 and BTV-MNG3/2016. For Segment 6, identities for the classical serotypes varied from 56.9% for BTV-15 up to 72.3% for BTV-11. Here, the identities for the atypical BTV serotypes varied from 68.2% for BTV-26 to the highest identity for BTV-25-TOV with 93.6%. In comparison, Segment 10, a more conserved BTV segment, showed identities of 75.7% for BTV-18 up to 79.2% for BTV-3 with the classical BTV 1-24 and from 78.7% for BTV-27_v01 up to 94.4% for BTV-25-TOV in comparison with atypical BTV. This high similarity might be explainable by the reassortment of BTV-36-CH2019 with BTV-TOV.

The BLAST results of the nucleotide and amino acid sequences of the complete coding sequence of the BTV-36-CH2019 segments are shown in Table 2. The most strongly related strains for all segments were found to be nearly solitary representatives of atypical BTVs, which is consistent with the phylogenetic trees. Interestingly, the sequence identities for Segment 2 are in both nt and aa, the most deviated ones towards other BTV strains with less than 67% identity. Furthermore, the most strongly resembling sequences towards Segment 2 of the Swiss BTV-36-CH2019 in nt and aa were reported to occur in Asia (BTV-XJ1407, BTV-MNG3-2016, BTV-MNG2-2016, V196-XJ-2014) or Africa (BTV-Y-TUN2017). In all other segments, BTV-36-CH2019 matches with the Swiss BTV-25-TOV strain and the French atypical serotype 27. 

For the atypical BTV strains, “serotyping” based on molecular data with defined criteria and therefore classification into “putative novel serotypes” is a practical solution [3,23]. Minimum levels of Seg-2/VP2 sequence identities for members of the same serotype were defined as 68.4% (nt)/72.6% (aa) [23]. Segment 2 of BTV-36-CH2019 matched to 66.83% on nucleotide level with BTV-XJ1407 and 63.39% on aa level with BTV-MNG3/2016. Based on the sequence data analysis it can be concluded that BTV-36-CH2019 forms a putative member of the novel atypical serotype 36. 

## 4. Discussion

The virus strains BTV-36-CH2019 and BTV-25-CH2019-1 were detected in clinically diseased goats on a Swiss alpine pasture. BTV-36-CH2019 represents the first member of the putative novel atypical BTV serotype 36, further expanding the group of atypical BTV strains, whereas BTV-25-CH-2019 is highly related to BTV-25-GER20018 and TOV. Up to now, 27 serotypes have been reported [24,35]; however, when novel atypical BTV serotypes are considered, 35 were reported [3]. Due to the results of the intense diagnostic clarifications and most of all due to the results of the experimental inoculation of goats, it is very unlikely that BTV-36-CH2019 was the causative agent of the clinical signs observed in the field-infected goats. Throughout the animal trial with BTV-36-CH2019, all goats remained clinically healthy. Nevertheless, goats are susceptible to BTV-36-CH2019 with a productive virus infection. During the 42 days of the animal trial, in four of the five goats inoculated subcutaneously with BTV-36-CH2019, viremia was detected, but neither clinical signs nor even a rise in rectal temperature were observed. 

Nevertheless, the experimental infection showed the general susceptibly of goats to BTV-36-CH2019. These findings are in line with reports about other atypical BTV strains, which are known to cause no clinical disease, such as BTV-25-TOV, BTV-26 or BTV-27 [22,36,37]. With reasonable certainty, the oedema and severe dermatitis observed in the field-infected goats of the Swiss pasture were not caused by the atypical BTV strain BTV-36-CH2019. A similar clinical picture of oedema of the ears and back with ulcerative, necrotizing, and exudative dermatitis in goats was reported after ingestion of *Froelichia humboldtiana* leading to primary photosensitization [38]. Another study revealed *Pastinaca sativa* as the cause of photosensitization via contact photodermatitis leading to severe necrotizing dermatitis in goats [39]. Recently, un-induced thermal burns leading to dorsal skin necrosis in Texel sheep were reported in Uruguay, particularly in obese sheep, sheep with dark skin or after sunlight exposure during hot months after shearing [40]. However, the evidence of solar-induced skin necrosis or photosensitization caused by plant contact or ingestion remains unproven for the goats of the Swiss pasture. 

An interesting finding with regard to the infection kinetics of BTV-36-CH2019 during the experimental infection was the late start of viremia with the first positive BTV genome detections on 10 dpi, followed by 12 dpi and even 17 dpi. This is in contrast to goat experiments performed with BTV-25-TOV, where the in total six infected goats developed viremia within 4 to 7 dpi [22,27], and to the experimental infection of goats with BTV-26, where viremia started 2 to 7 dpi with peak Cq values of 19 to 23 at 9–11 dpi (Batten et al. 2013). However, the results of our trial with BTV-36-CH2019 more strongly resemble the results of the experimental infection of goats with BTV-27, where viremia also started late after 7 dpi up to 14 dpi, with two infected goats remaining negative during the study period [37]. The infectious dose of 10^5.0^ CCID_50_/_mL_ used in our study represents a sufficient dose for BTV infection in comparison to other study designs. The host individual genetic background may explain the remaining negativity of one goat. The existence of BTV-resistant individual animals is stated, but further research is necessary [10]. As both control goats remained negative during the animal trial, no evidence of contact transmission is given for BTV-36-CH2019. Furthermore, the oral, nasal, ocular and rectal swabs were all negative for BTV genome except one, which makes the scenario of horizontal transmission via secretions or excretions by direct contact unlikely in the case of BTV-36-CH2019. This is in line with the findings for BTV-25-TOV, where no swab material was found positive either, but in contrast to BTV-26 with findings of positive nasal and ocular swabs [27,41]. Nevertheless, we detected one positive nasal swab of goat #14 on 21 dpi. We presume the positivity of the nasal swab was due to blood contamination during sampling and not a consequence of active virus shedding via nasal secretions. However, the horizontal transmission potential of BTV-36-CH2019 needs to be further analyzed, as the length of the current animal trial of 42 dpi together with the late viremia of 10-17dpi can be evaluated as a rather short period for transmission surveillance. In comparison, during the BTV-27 animal trial one contact goat was detected BTV genome positive starting on 56 dpi [37].

Spleen, lung and liver of the four goats with viremia were positive for BTV genomes in RT-qPCR, which is in line with other BTV infection studies [9,16,27,36]. The highest viral loads in the spleens are not surprising, as BTV is associated to red blood cells and the spleen represents the major blood reservoir in the body; however, only a small amount of virus replication has been reported for the spleen [7]. The largest amount of BTV regarding the lymphatic tissue was found in the tonsils and the head lymph nodes, but they are not known for strong BTV replication either [7]. In the study with BTV-25-TOV and BTV-26, viral genome was detected in the lymph node materials as well [27,36]. However, the viral spread to different organs for BTV-25-TOV was shown to be rather slow [27], which might explain the negative results for BTV genomes in the mesenteric lymph nodes of the BTV-36-CH2019-infected goats. Moreover, both skin samples of the four viraemic goats were negative, likewise observed for the skin sample of a BTV-25-TOV-infected goat on 8 dpi [27] Nevertheless, the endothelial cells of skin capillaries were identified to be an important organ for BTV replication, at least for the classical BTV serotype 2 [7]. Therefore, the viral transmission and pathogenesis of BTV-36-CH2019 and the atypical BTV serotypes in general need further research.

None of the goats infected with BTV-36-CH2019 developed group-specific BTV antibodies measurable in the cELISA or serotype-specific antibodies measurable in the VNT despite the presence of viremia in four of the goats. For BTV-25-TOV, BTV-27 and BTV-25-GER2018, the antibody levels were described in general as low and slowly increasing [27,29,37], which might be in line with the phylogenetically related BTV-36-CH2019. Hence, a longer time span post infection might be necessary for BTV-36-CH2019 infected goats to seroconvert. Furthermore, commercially available ELISA kits using the VP7 of classical BTV strains might not be an ideal tool for measuring antibodies formed against BTV-36-CH2019. Several BTV-36-CH2019 field-infected goats were positive in ELISA and consequently might have formed antibodies against BTV-36-CH2019. However, the virus neutralization test performed with the BTV-36-CH2019 positive rabbit serum and the experimentally infected goats led to incomplete neutralization as already described for BTV-25-GER2018 [29] and BTV-MNG1/2018, BTV-MNG2/2016, and BTV-MNG3/2016. Interestingly, we observed a cross-reactivity in virus neutralization with the serum reactive to BTV-MNG1/2018. The VNT targets antibodies formed against the VP2 of BTV, which is encoded by Segment 2. Cross-neutralization in VNT or SNT is reported for closely related BTV strains; however, phylogenetically, the BTV-36-CH2019 strain is quite distant to its nearest neighbors in the nt and aa of Segment 2. The phylogenetic analysis of the nt of Seg-2 revealed the highest sequence identities with BTV-XJ1407 with 66.8%, followed by BTV-MNG1/2018 with 66.6%, whereas the closest neighbor in aa of the VP2 is BTV-MNG3/2016 with 63.4%. BTV-MNG1/2018 is at least amongst the nearest neighbors of BTV-36-CH2019 (62.1% in aa), which, however, cannot sufficiently explain the cross-neutralization in VNT.

Phylogenetic analysis revealed the nearest neighbors of BTV-36-CH2019 in the nt and aa of all segments (except the Segment 2/VP2) amongst the novel BTV atypical serotypes 25 and 27, particularly with the Swiss BTV-25-TOV. The geographical occurrence of BTV-25 and BTV-27 strains was reported in Italy, Germany, France and Switzerland (TOV), thus in close proximity to the occurrence of BTV-36-CH2019. BTV has in general a very high genetic diversity even within the same serotype. Furthermore, the founder effect could lead to the development of geographic topotype variations within the same serotype [42]. However, Segment 2 of BTV-36-CH2019 might have an Asian origin as it resembles the Chinese BTV-XJ1470 or the three Mongolian strains with high identities. The global spread of BTV is a complex, multifactorial development driven by environmental and anthropogenic factors [43], and reassortment events are not limited to phylogenetically related viruses [44] and might have also occurred in the case of BTV-36-CH2019. 

BTV-25-CH20019-1 was found to circulate geographically very closely to the location of Toggenburg Virus. Hence, it is not surprising that in all 10 segments the molecular proximity to TOV, but also BTV-25-GER2018, can be seen. Interestingly, a certain number of goats were co-infected with BTV-25-CH2019-1 and BTV-36-CH2019. Even though BTV-36-CH2019 and BTV-25-CH20019 are related in 9 of 10 segments, the serotype-defining segment 2/VP2 is molecularly quite different and explains the missing cross-protection in the field-infected goats. The four BTV-positive goats from Flocks C and D were positive for BTV-25 RNA only. Furthermore, in Flocks C and D, no sudden clinical signs were reported. Thus, it is very unlikely that the clinical signs reported only in Flock A were caused by BTV-25-CH-2019-1. This is in line with the findings of the closely related TOV and BTV-25-GER2018 strains, which are asymptomatic in goats as well [22].

In conclusion, BTV-36-CH2019 was discovered in diseased goats in Switzerland and expands the group of atypical BTV forming the putative novel atypical serotype 36, whereas the novel BTV-25-CH2019-1 strain is part of serotype 25. Our study clearly indicates that BTV-36-CH2019 was not the causative agent of the clinical disease in the field-infected goats, as none of the experimentally infected goats showed any sign of disease or fever. Our finding is in line with previous studies on other atypical BTV serotypes and strains also not resulting in clinical disease. However, the immune response, the susceptibility of further ruminant species and the transmission route of BTV-36-CH2019 need further research to better understand their epidemiological role.

## Figures and Tables

**Figure 1 viruses-13-00721-f001:**
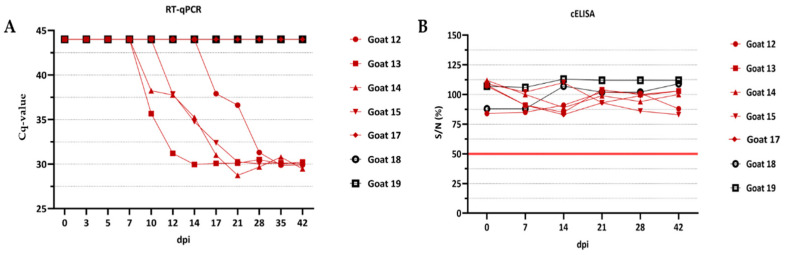
Overview of the RT-qPCR and cELISA results of the experimental infection of 5 goats. The Cq values of the Pan-BTV-S10 RTq-PCR are shown in (**A**) and considered positive, when Cq < 40. The reactivities of the ID-vet cELISA in percent of negative control (≥50% is negative according to the manufacturer) are shown in (**B**). The red line represents the cut-off.

**Figure 2 viruses-13-00721-f002:**
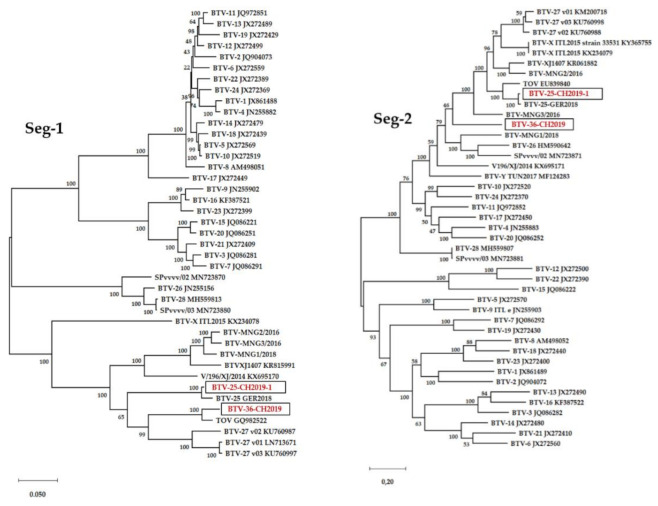
Phylogenetic analyses of full-genomes of strains BTV-36-CH2019 and BTV-25-CH2019-1. The phylogenetic trees of each of the 10 segments were created with MegaX using the genetic distinction model Tamura-Nei and tree-built method maximum likelihood including BTV strains representing the known BTV serotypes published [26,34]. We performed a bootstrap analysis with 100 replications. The black boxes surround the BTV-36-CH2019 and BTV-25-CH2019-1 sequences highlighted in red.

**Table 1 viruses-13-00721-t001:** Overview of the RT-qPCR results of the organ samples on 46 dpi and the EDTA blood on 42 dpi of the experimental goat inoculation. A sample was considered positive in Pan-BTV-S10 RT-qPCR_V2, when Cq < 40, whereas ‘-’ means no Cq. No pathological abnormalities were observed.

		Organ Material					
Goat ID	EDTA Blood	Skin 1	Skin 2	Lung	Liver	Spleen	Mesenteric Lymph Node
#12	29.9	-	-	32.3	32.1	29.2	-
#13	30.23	-	-	34.4	32.4	30.7	-
#14	29.47	-	-	33.2	31.1	33.1	-
#15	30.05	-	-	32.3	31.7	31.4	-
#17	-	-	-	-	-	-	-
#18	-	-	-	-	-	-	-
#19	-	-	-	-	-	-	-

**Table 2 viruses-13-00721-t002:** Closest neighbors of BTV-36-CH2019 (CDS) following BLAST analyses. The three highest identities/similarities with a query coverage of >85% are presented.

Segment/Protein (Accesion No.)	Serotype (nt/aa)	Strain (nt/aa)	Accession No. (nt/aa)	Identity Level % (nt/aa)	Query Cover %
	25/25	TOV/TOV	GQ982522.1/ACY02806.1	96.52/98.23	100/99
1/VP1 (LR993239)	27/unknown	BTV-27-FRA2014-v02/BTV-MNG3-2016	KU760987.1/CAD2286107.1	85.21/93.55	100/99
	27/unknown	BTV-27-FRA2014-v01/BTV-MNG1-2018	LN713671.1/CAD2286086.1	85.01/93.24	100/99
	Unknown/unknown	BTV-XJ1407/BTV-MNG3-2016	KR061882.1/CAD2286108.1	66.83/63.39	86/99
2/VP2 (LR993240)	Unknown/unknown	BTV-MNG3-2016/BTV-Y-TUN2017	LR877359.1/AVQ09328.1	66.34/63.19	95/96
	Unknown/unknown	BTV-MNG2-2016/V196-XJ-2014	LR877348.1/ASW41947.1	66.14/62.41	99/99
	25/25	TOV/TOV	GQ982523.1/ACY02807.1	97.93/99.11	100/99
3/VP3 (LR993241)	27/unknown	BTV-27-FRA2014-v03/BTV-MNG3-2016	KU760999.1/CAD2286109.1	84.85/95.34	100/99
	27/unknown	BTV-27-FRA2014-v02/BTV-MNG2-2016	KU760989.1/CAD2286091.1	84.66/95.34	100/99
	25/25	TOV/TOV	GQ982524.1/ACY02808.1	97.99/98.60	100/99
4/VP4 (LR993242)	Unknown/unknown	BTV-Z ITL2017/BTV-Z ITL2017	MF673723.2/AVA16291.2	90.63/97.27	91/91
	Unknown/unknown	BTV-25-GER2018/BTV-MNG3-2016	LR798444.1/CAD2286110.1	90.59/92.24	100/99
	25/25	TOV/TOV	EU839841.1/ACJ06703.1	97.33/96.72	99/99
5/NS1 (LR993243)	Unknown/unknown	BTV-25-GER2018/BTV-25-GER2018	LR798445.1/CAB5237905.1	78.34/83.24	99/99
	4/11	MOR2004-02/USA2013-WA 13-031503	KP821423.1/AKM21154.1	75.17/79.60	99/99
	25/25	TOV/TOV	EU839842.1/ACJ06704.1	96.58/97.15	100/99
6/VP5 (LR993244)	Unknown/27	BTV-25-GER2018/BTV-27-FRA2014-v01	LR798446.1/CEK41875.1	82.86/86.69	100/99
	27/unknown	BTV-27-FRA2014-v01/BTV-28-1537-14	LN713675.1/QDH76491.1	77.74/84.79	100/99
	25/25	TOV/TOV	EU839843.1/ACJ06705.1	97.52/99.71	100/99
7/VP7 (LR993245)	27/26	BTV-27-FRA2014-v03/KUW2010-02	KU760993.1/AED99449.1	84.48/97.99	100/99
	27/unknown	BTV-27-FRA2014-v02/Spvvvv-03	KU761003.1/QGW56811.1	84.19/97.71	100/99
	25/25	TOV/TOV	EU839844.1/ACJ06706.1	97.46/97.45	100/99
8/NS2 (LR993246)	25/27	BTV-25-GER2018/BTV-27-FRA2014-v03	LR798448.1/AMQ36834.1	96.61/85.84	100/99
	27/27	BTV-27-FRA2014-v03/BTV-27-FRA2014-v01	KU761004.1/CEK41877.1	82.67/85.55	100/99
9/VP6 (LR993247)	25/25	TOV/TOV	EU839845.1/ACJ06707.1	97.17/*96.35*	99/99
	Unknown/unknown	BTV-25-GER2018/BTV-25-GER2018	LR798449.1/CAB5237909.1	85.96/*83.59*	99/99
	Unknown/unknown	BTV-MNG1-2018/BTV-MNG1-2018	LR877345.1/CAD2286102.1	81.89/*79.03*	99/99
10/NS3 (LR993248)	25/27	TOV/BTV-27-FRA2014-v02	EU839846.1/AMQ36826.1	94.64/95.20	100/99
	Unknown/25	BTV-25-GER2018/TOV	LR798450.1/ACJ06708.1	87.25/93.89	100/99
	27/unknown	BTV-27-FRA2014-v02/BTV-X ITL2015	KU760996.1/APC23697.2	86.96/93.01	100/99

## Data Availability

All data presented in this study are summarized in the paper. The detailed data of this study are available on request from the corresponding author.

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
