# Peer review of "Putative Novel Atypical BTV Serotype ‘36’ Identified in Small Ruminants in Switzerland"

_viruses, 2021, doi:10.3390/v13050721_

Round 1

Reviewer 1 Report

Manuscript title: Putative novel atypical BTV serotype ‘36’ identified

3 in small ruminants in Switzerland.

Comments

In this manuscript by Ries et al., the putative atypical BTV serotype "36" was found in a Swiss herd of goats. Clinical signs included multifocal purulent dermatitis, facial edema, and fever. After detecting BTV by RT-qPCR, serological typing identified BTV-25 and some new BTV serotypes in the affected goats. However, among experimentally infected goats, no fever or clinical disease was observed in any goats that had been vaccinated. Four goats developed BTV viremia, and one vaccinated goat and two contact animals remained negative. Since no viral RNA was detected in the swab sample and the clinical signs in the initial flock were probably not due to infection by the new BTV-36-CH0219 serotype, the whole experiment remains unclear and inconclusive.

Some specific comments are listed below:

  1. Line 78: “an only low” should be changed to “a low”.
  2. Although the authors have expended considerable effort to validate the role of the two atypical virus strains, BTV-36-CH2019 and BTV-25-CH2019-1, only viremia and virus particles in blood could be detected, and neither clinical signs nor immune responses were observed. Perhaps selecting a more susceptible sheep breed for these experiments would have been more appropriate. Alternatively, these pathogens may not be responsible for triggering the observed symptoms?
  3. Based on the above-described points, these two isolates failed Koch's hypothesis for verification. Could the serious clinical signs observed in the initial goat flock be caused by other pathogens or are those atypical isolates exerting synergistic effects resulting in symptomology?
  4. Although no fever or clinical disease has been observed in any goats that have been vaccinated, this is still a well-designed set of studies aimed at the infectivity of these viruses.

Reviewer 2 Report

This is a well-organized and communicated study and I have just a few minor comments for the authors to consider.

Minor English/grammatical revisions for improvement are included.

Throughout, where the authors use “could be” before declaring a result, it is best to use “was” or make the verb past tense.  “We identified” (abstract), “was isolated”, “was tested”, “was observed” (results)

Throughout, use either “on day x” or “at x dpi”

Throughout, cytopathic effect acronym is CPE (all capitals)

Throughout, bluetongue should not be capitalized unless starting the sentence.

Throughout,” x were positive” or” x tested positive” not “x were tested positive”.

Intro, the background provided for BTV8 seems a bit extensive for research regarding atypical BTV serotypes.

Line 60 needs a comma after change

Line 73 Clinical-pathological not clinic-pathological

Line 78 replace “…an only low…” with “…a low…”

Line 93 replace “…symptoms…” with “…signs…”

Line 103 remove “a” before clinically (since herds is plural)

Line 110 "enhanced" (past tense)

2.1.1. In the description of this flock it should be indicated that this is the flock that was clinically affected.  It makes more sense then that B, C, and D were tested for surveillance to inform what was happening in flock A.  In describing this flock, how many of the 77 animals were showing signs of disease?  “None of the animals were vaccinated…”

2.1.2. A comma should follow “breeds”. “For three goats, no breed information was available.” None of the animals were vaccinated…”

2.1.3. Were the two samples sent to IVI from clinically infected animals? Explain “investigation”. What exactly?  Explain what further analysis was to be done at FLI. Provide the acronym for FLI here at first use. How were the five goats chosen for euthanasia? What further analysis was to be done for the samples sent to IVP? 

2.2. IVI already defined in line 134.  FLI should be defined in line 137 not here.  Which organs? Explain ‘All’.  Swabs from what? Explain ‘diverse’. Line 181 comma should follow strain. Lines 182-3, what is meant by 50 and 30?  5’ and 3’ ?

2.3. The section indicates 8 blood samples were chosen for triple passage to isolate BTV-36 and 4 blood samples were triple passaged for BTV-25 isolation.  How many of these isolates were used for sequencing? How were the Cq value ranges for each virus type determined? Previous work or is there a reference? And what criteria were used to choose those isolates/passages that would then be used for sequencing? Which ones were chosen? And at what passage were they used? Were additional passages not done if CPE was observed on the first or second passage? Most importantly, the authors do not explain, here or in results, how they accounted for sequence changes that may have occurred during the triple passages of these goat isolates in hamster cells or whether sequencing attempts without cell culture passages were considered/attempted.

2.4.2. line 235-6 could be written more clearly. ‘...negativity compared to the negative control’ is confusing

2.4.3 Write out ‘five’

2.5. Where (anatomically) were the goats inoculated? Four should be written out. Four passages are not mentioned in the methods.

2.6. Sequences were generated from which passages of which samples? Line 265 replace “earlier” with “previously” and it is unnecessary to indicate that the cDNA was double stranded.  In my opinion. Define aa and nt at first use. Suggest rewording this sentence. “Furthermore, the consensus sequences of each of the 10 segments of BTV-36-CH2019 were determined by BLAST (NCBI) analysis against the nucleotide/amino acid (nt/aa) databases to identify the nearest molecular neighbours.”

3.1.1 “oedema” or “edema”. Be consistent.  Replace “from two animals…” with “for two animals…”.

3.1.2. suggest rewording. “Macroscopically, severe skin alterations could be seen on the ears, the bridge of the nose, the shoulder blades, and along the spine.”  “…alopecic with severe crusting…”  “…necrotic and had started to peel.”

3.2 This is difficult to follow as written. It could be improved to more clearly state what samples from what flocks were positive for which virus. 

3.3. Three passages are referred to (line 319) but inoculum used for experimental goats was four at the titer indicated in line 316.

3.4.2.  The ELISA numbers don’t make sense. Clarify which ELISA in line 330.  If ELISA and not cELISA, then numbers in this sentence contradict previous sentence.

3.5. “No clinical signs or fever…” Spaces are missing between day number and dpi throughout this section.

Figure 1 legend. ID-Vet not ID.vet

Line 396, BLAST
